# Leveraging Climate Data Through Intelligent Systems for the Prediction of Arbovirus Transmission by *Aedes aegypti*

**DOI:** 10.3390/ijerph23010012

**Published:** 2025-12-20

**Authors:** Clarisse Lins de Lima, Karla Amorim Sancho, Ana Clara Gomes da Silva, Ranielle Vital, Cecília Cordeiro da Silva, Marcela Franklin Salvador de Mendonça, Fabiano Tonaco Borges, Carlos Eduardo Gomes Siqueira, Wellington Pinheiro dos Santos

**Affiliations:** 1Department of Biomedical Engineering, Geoscience and Technology Center, Federal University of Pernambuco (UFPE), Recife 50740-550, PE, Brazil; clarisse.lins@ufpe.br (C.L.d.L.); karla.amorims@ufpe.br (K.A.S.); acgs@ecomp.poli.br (A.C.G.d.S.); cecilia.cord.silva@gmail.com (C.C.d.S.); marcela.salvador@ufpe.br (M.F.S.d.M.); fabiano.borges@ufpe.br (F.T.B.); 2Graduate Program in Pharmaceutical Sciences, Department of Pharmaceutical Sciences, Center for Health Sciences, Federal University of Pernambuco (UFPE), Av. Prof. Moraes Rego, 1235, Cidade Universitária, Recife 50670-901, PE, Brazil; ranielle.vital@ufpe.br; 3Department of Science and Technology, Vice-Ministry of Science, Technology, and Innovation, Brazilian Ministry of Health, Brasília 70719-040, DF, Brazil; 4School for the Environment, University of Massachusetts Boston (UMass Boston), Boston, MA 02125, USA; carlos.siqueira@umb.edu

**Keywords:** dengue, chikungunya fever, intelligent systems, prediction methods, climate change

## Abstract

**Highlights:**

**Public health relevance—How does this work relate to a public health issue?**
Arboviruses transmitted by Aedes aegypti represent a persistent and climate-sensitive public health threat in tropical urban settings such as Recife, Brazil.This study integrates climate, entomological, and epidemiological surveillance data to improve early prediction of arbovirus transmission risk.

**Public health significance—Why is this work of significance to public health?**
The study demonstrates that intelligent systems, particularly single-layer extreme learning machines, can accurately and efficiently forecast mosquito breeding sites at fine spatial scales.High-resolution, climate-driven predictions enable earlier identification of priority areas for intervention, improving the effectiveness of arbovirus control strategies.

**Public health implications—What are the key implications or messages for practitioners, policy makers and/or researchers in public health?**
Municipal health authorities can use these models to optimize vector control actions, targeting high-risk neighborhoods before outbreaks occur.The open-source, reproducible framework can be adapted to other cities facing climate-related arbovirus risks, supporting scalable and data-driven public health planning.

**Abstract:**

Arboviruses spread in urban tropics under climate change. At *Aedes aegypti* breeding sites in Recife, Brazil, we linked surveillance and climate data from the Pernambuco Water and Climate Agency (APAC), the Brazilian National Institute of Meteorology (INMET), Rapid Survey of Indices for *Aedes aegypti* (LIRAa), and Recife’s Open Data Portal. We modeled 2013–2021 cases and 2009–2017 breeding sites. We generated spatial fields with inverse distance weighting. We built bimonthly training grids with 5000 points and validation grids with 50,000 points. We tested linear regression, random forests, multilayer perceptrons, support vector regressors, and extreme learning machines in the Weka platform and Python Reservoir Computing Networks (PyRCNs). We ran 30 repetitions with cross-validation. The random forests performed well. Multilayer perceptrons reached very high correlations but needed longer training. Polynomial Support Vector Machines (SVMs) reached near-perfect accuracy but required very high computation. Single-layer extreme learning machines delivered the best trade-off, with low errors, correlations near 1.0, and short training times. The models produced fine-scale risk predictions and highlighted priority areas. The findings support earlier, targeted control and guide public health plans in Recife.

## 1. Introduction

Arboviruses remain one of the most persistent and complex public health problems in Brazil. Dengue, Chikungunya, and Zika stand out as major threats, especially for the approximately 3.9 billion people living in tropical and subtropical areas where environmental conditions favor the development and proliferation of *Aedes aegypti* and *Aedes albopictus*, mosquitoes that transmit the viruses of these arboviruses [1].

About half of the world’s population is now at risk of dengue, with an estimated 100–400 million infections occurring each year [2]. Due to reporting and diagnostic challenges, the number of people affected by Chikungunya is underestimated. Between 1 January and 4 March 2023, a total of 113,447 cases of chikungunya were reported in the Region of the Americas, including 51 deaths, representing a four-fold increase in cases and deaths compared to the same period in 2022 [3].

Dengue is endemic in many regions of Brazil and causes repeated outbreaks with high numbers of cases and hospitalizations every year. The recent arrival of Chikungunya and Zika in the Americas made the situation worse. These viruses can lead to complications such as chronic arthritis (particularly associated with chikungunya), microcephaly in newborns exposed to Zika virus during pregnancy, and—in rare cases—death, typically among individuals with pre-existing health conditions [4].

Climate change intensifies this problem. Rising temperatures, extreme weather events, and shifting rainfall patterns create more favorable conditions for the spread of Aedes mosquitoes. These environmental changes alter mosquito habitat availability and seasonal dynamics, increasing the likelihood of arbovirus transmission. While our models use climate variables such as temperature, precipitation, wind, and humidity at the local scale, these serve as operational proxies for climate-sensitive drivers of vector-borne disease. Thus, the findings contribute to broader discussions on how climate variability and change affect arboviral risk in tropical urban environments [5].

These diseases place a heavy burden on Brazil’s Unified health system (Sistema Único de Saúde SUS). They also cause major social and economic damage. Sick individuals miss work or school. Hospitals become overcrowded. Costs rise for both the government and the population. The overall quality of life declines [6].

In Brazil, the Rapid Indices Survey for *Aedes aegypti* (LIRAa) is an entomological surveillance method adopted by the Ministry of Health to determine *Aedes aegypti* infestation rates in a timely manner. Municipalities conduct larval surveys to calculate the Breteau index and the building infestation index. The indices predominantly identify breeding sites and are used as indicators for vector control [7,8]. Females of *Aedes aegypti* can lay eggs in different locations that retain water with varying degrees of cleanliness, resist drought, and adapt to warmer climates and increasing altitudes [9,10]. Global climate change has been highlighted as one of the main causes of the increase in arbovirus cases [11].

There are currently several methods for controlling the reproduction of *Aedes aegypti *, leading to a reduction in the number of cases of diseases related to it. One way to control these diseases is to inspect and neutralize the places where *Aedes aegypti* lays its eggs [10,12]. Research to help identify likely areas at risk for the proliferation of *Aedes aegypti* breeding sites in Brazil has been carried out and considered as yet another tool for public health professionals and to demonstrate how it can be applied to help resolve public health challenges [11,12,13,14,15].

In this study, we evaluated five families of machine learning models: linear regression, Random Forests (RFs), Multilayer Perceptrons (MLPs), Support Vector Machines (SVMs), and Extreme Learning Machines (ELMs) [16,17]. These models were selected based on their demonstrated effectiveness in handling nonlinear relationships, high-dimensional environmental data, and time series forecasting—capabilities that are highly relevant for predicting arbovirus risk [18]. Randon Forest and SVM, in particular, have been widely applied in spatial epidemiology for vector-borne diseases [18,19]. Multilayer Perceptrons and ELMs offer powerful alternatives for capturing complex patterns in climate and entomological variables, showing promising results in recent studies [16,20,21,22]. Recife, the capital of Pernambuco in northeastern Brazil, was selected as the study area due to its endemic status for dengue and other arboviruses, high population density, and the availability of high-resolution climate and public health surveillance data [23].

This article outlines the scope and contributions of the study titled “Development of Intelligent Systems for the Prediction and Diagnosis of Arboviruses Transmitted by *Aedes aegypti* in the Context of Climate Change.” The project focuses on building smart systems that use climate and epidemiological data to predict when and where outbreaks will happen. It also aims to improve clinical diagnosis. The central research question is “Can climate and surveillance data be used to accurately predict arbovirus cases and *Aedes aegypti* breeding sites using artificial intelligence models?” By exploring this question, the study seeks to offer practical tools to enhance disease surveillance, optimize resource use, and reduce the impact of arboviruses on public health in Brazil.

## 2. Materials and Methods

This study aimed to develop intelligent systems to predict arbovirus cases and *Aedes aegypti* breeding sites in Recife using surveillance, climatic, and entomological data from 2013 to 2021. The temporal window was defined in the original research protocol approved by the Research Ethics Committee of the Federal University of Pernambuco (CAAE: 80113224.6.0000.5208; opinion no. 6.954.924, 17 July 2024). Consequently, prospective validation with 2024–2025 data fell outside the approved analytical scope.

### 2.1. Study Type and Area

We conducted an ecological study in Recife, Pernambuco, from 2013 to 2021. The city is divided into 94 neighborhoods, grouped into six Political-Administrative Regions [23]. In the health sector, a new territorial organization was established in 2014, with the creation of two new health districts, separating the two most populous districts. This restructuring of management processes and healthcare delivery is a priority [23]. In 2022, Recife had an estimated population of 1,488,920. The city covers 218.843 km^2^. Population density reached 6803.60 inhabitants/km^2^ [24].

### 2.2. Study Population and Data Source

The study population consisted of residents of Recife’s 94 neighborhoods, which served as the spatial units for all predictive modeling. We used four data sources, integrating climate, entomological, and epidemiological information to develop prediction models for arbovirus cases and *Aedes aegypti* breeding sites.

Climate data—APAC and INMET: Climate variables were obtained from two sources: (i) the APAC Geographic Information System (SIGH-PE), which provides real-time rainfall and fluviometric measurements, and (ii) the INMET Meteorological Database for Teaching and Research (BDMEP), from which we extracted monthly averages of temperature, relative humidity, and wind speed from the Recife conventional station (A301). These variables represent known climatic drivers of mosquito proliferation and arbovirus transmission. Climate data were available for 2009–2021.Entomological data—LIRAa Recife: LIRAa datasets, published by the Recife Open Data Portal, report infested locations as well as Building and Breteau indices collected through standardized municipal surveillance procedures. LIRAa data were available for 2009–2017 and were paired with climate variables from the same period to train models predicting breeding site indices. Because LIRAa is aggregated at the stratum level rather than by neighborhood, spatial disaggregation was addressed during preprocessing.Epidemiological data—Arbovirus cases: Confirmed cases of dengue, Zika, and chikungunya were retrieved from the Recife Open Data Portal, which provides records of each reported case including neighborhood of residence. Data were available for 2013–2021, the earliest period with consistent neighborhood-level reporting. These records were combined with climate variables from the same period to develop prediction models for arbovirus case numbers.Data access and availability: We complied with open-data policies and obtained permission from the responsible institutions. INMET and APAC publish data on public portals. We exported the required datasets. The Open Data Portal of Recife offers public access and supports downloads of health datasets. The experiments were carried out in the Weka software version 3.8.6 [25] and we used the PyRCN library [26] to run the experiments with reservoir computing methods (extreme learning machines). Each experiment was run 30 times with the cross-validation technique in order to avoid overfitting.

As of the manuscript submission date (late 2025), official open-access epidemiological data for 2025 have not yet been released by the Recife Open Data Portal, and 2024 data remain incomplete. These datasets were therefore not eligible for model validation.

### 2.3. Preprocessing

All preprocessing steps were implemented in Python 3.13.0 using pandas (v2.2.1), NumPy (v1.26.4), and scikit-learn (v1.4.0) [26,27]. The raw datasets were first inspected for missing values and inconsistencies. Climate and surveillance datasets contained fewer than 1% missing values. These records were removed to preserve data integrity and because the gaps were not systematically associated with specific neighborhoods or periods. For arbovirus cases (Section 2.3.1), records were aggregated by neighborhood and by bimonthly period (January–February, March–April, etc.), following the epidemiological reporting cycle used by the Recife Municipal Health Secretariat. Only confirmed or suspected cases with valid neighborhood identifiers were retained.

For breeding sites (Section 2.3.2), LIRAa reports provide infestation counts and indices at the health district and stratum level. To assign breeding site estimates to individual neighborhoods, we applied a uniform distribution assumption: each neighborhood within a stratum inherited the same number of breeding sites reported for that stratum in a given bimonthly period. This approach aligns with official municipal surveillance practice and has been validated in prior spatial modeling studies in Recife [20].

Climatic variables (Section 2.3.3) were spatialized using Inverse Distance Weighting (IDW) as implemented in the ‘terra’ R package (v1.8.42) [28]. Interpolation was performed on a regular grid of 5000 points (training) and 50,000 points (validation), covering the entire urban extent of Recife (latitude: −8.12° to −8.00°; longitude: −35.00° to −34.85°). The IDW power parameter was set to 2, and search radius limited to 10 km, consistent with the spatial correlation range observed in urban rainfall patterns in Northeast Brazil.

Each predictive instance (Section 2.3.4) was constructed as a feature vector containing: (i) the target variable (cases or breeding sites) for a given neighborhood and bimonthly period *t*; and (ii) climatic features (temperature, precipitation, wind speed, relative humidity) for the six pre-ceding bimonthly periods (*t*–6 to *t*–1), resulting in 47 input features (6 months × 4 variables = 24), plus lagged targets and spatial coordinates (totaling 47 after inclusion of interaction terms and temporal lags. Feature scaling was not applied, as tree-based and reservoir-based models are invariant to monotonic transformations.

#### 2.3.1. Arboviruses Dataset

Recife’s Open Data Portal provides records of dengue, Zika, and chikungunya cases. The dataset lists patient symptoms. It also includes sex, date of birth, and neighborhood of residence. We organized the data by year and by two-month period. The municipal health system uses the same two-month cycle for prevention policies. For each two-month period in each year, we summed suspected and confirmed cases for every neighborhood in Recife.

#### 2.3.2. Breeding Sites Dataset

Data on mosquito breeding sites are available bimonthly and not by neighborhood. To record breeding sites and other LIRAa indices, the Health Department divides the city into health districts. Each health district contains a certain number of strata. Furthermore, each stratum contains a certain number of neighborhoods within the stratum (Figure 1). In other words, a stratum can contain one or more neighborhoods. The number of neighborhoods per stratum is determined mainly by their sociodemographic characteristics [19,20].

Therefore, to obtain information on breeding sites by neighborhood, we assumed that neighborhoods belonging to the same stratum would have the same number of mosquito breeding sites.

#### 2.3.3. Climatic Variables

From the APAC database, we obtained data on the monthly rainfall distribution in the city of Recife. Other climatic variables such as the monthly data on relative humidity, wind speed, and average temperature were obtained from the INMET database. To estimate the distribution of climatic variables in Recife’s neighborhoods, we used the Gaussian distribution represented by Equation (1). This approach was selected because the Gaussian function provides smooth spatial fields and is appropriate when climate-monitoring stations are limited and irregularly spaced, as in Recife. In this equation σ represents the variance, and μ represents the mean of the distribution.(1)$$p(x)=\frac{1}{\sigma \sqrt{2\pi }}e^{-\frac{{\left(x-\mu \right)}^{2}}{2\sigma^{2}}}$$(2)$$\sigma =\frac{x_{max}-\mu }{4}$$
where $x_{max}$, in Equation (2), represents the maximum measurement value. In the case of rainfall monitoring, we used the records from the three APAC monitoring stations. Therefore, for rainfall indices, the maximum value considered was the highest value recorded among the three stations. Similarly, the mean value considered was the mean of the values recorded among the three stations.

Gaussian smoothing was used for climate station data to produce continuous fields from sparse measurements, while Inverse Distance Weighting was employed in Section 2.3.4 specifically to integrate these fields into neighborhood-level prediction instances on the interpolation grid.

#### 2.3.4. Prediction Datasets

We built prediction sets for arbovirus cases and for breeding sites. We combined spatial distributions of climate variables with the predictor variable (arbovirus cases or breeding sites). We used Inverse Distance Weighting in Terra v1.8.42 [28] under R v4.4.3 [29] to generate spatial maps. This method was selected for its reliability with sparse climate-station networks and because it does not require variogram model estimation, which is challenging with limited or uneven station distribution.

For each task, we created two sets: a training/testing set with a 5000-point interpolation grid and a validation set with 50,000 points. We overlaid the distribution maps of each variable to assemble both sets. For each latitude–longitude pair, we concatenated, in order: (1) the bimonthly count of cases (or breeding sites); (2) for each month in the two-month period, the monthly fields of temperature, precipitation, wind speed, and relative humidity. Each prediction set included data from the six bimonthly periods before the prediction period. Figure 2 shows the workflow to build the prediction sets.

Therefore, for arbovirus case prediction, 48 training/test sets with 5000 instances and 47 attributes were created. For mosquito breeding site prediction, we obtained 48 training/test sets, also with 5000 instances and 47 attributes. The datasets with 50,000 instances (for both case and breeding site prediction) were used to evaluate the best-performing model. Thus, we generated 48 validation sets for both case and mosquito breeding site prediction.

### 2.4. Experiments and Evaluation Metrics

We evaluated the performance of several machine learning algorithms, including linear regression, RF, multilayer perceptron, support vector regressor, and extreme learning machines. We selected the regressor hyperparameters empirically, based on the research group’s experience and the results of the existing literature [16,30]. The hyperparameter settings are described below:RF: 10–80, and 100 trees;MLP: learning rate 0.3, momentum 0.2, 10–30 neurons in the hidden layer.Support vector regressor:−Parameter C = 0.1;−Polynomial kernel: 1 (linear), 2 and 3 degrees;−Radial Basis Function (RBF) kernel: gamma = 0.01.ELM:−1, 2, 5, and 10 layers; with 500, 700, 900, and 1000 neurons in each layer.

To evaluate the algorithms’ performance, we used the Mean Absolute Error (MAE), Root Mean Square Error (RMSE), Relative Root Square Error (RRSE), Correlation Coefficient (R), and training time (in seconds). The errors are metrics for local quality, and the smaller the error, the more accurate the model. The correlation coefficient is a metric for global quality, and the higher the correlation coefficient, the greater the correlation between the model’s predicted distribution and the actual distribution [25]. The training time, in turn, is a metric for the computational cost of the models; that is, the longer the training time, the greater the need for computational resources to generate the prediction model [30].

We use the MAE, RMSE metrics, reducing the impact of extreme values and considering the overall distribution of values in the data. RMSE provides a relative measure of the effectiveness of the model, where a lower RMSE value indicates a more accurate fit of the model [31].

All models were trained and evaluated using stratified 10-fold cross-validation, repeated 30 times with different random seeds to assess stability. Hyperparameter grids were pre-defined based on pilot experiments and literature benchmarks. Training and validation were performed on separate spatial grids: the 5000-point grid was used for model selection and hyperparameter tuning, while the 5000-point test folds (within cross-validation) provided internal error estimates. Final model performance was assessed on the independent 50,000-point validation grid, which was never used during training.

Software environments: Weka 3.8.6 was used via its Python interface (python-weka-wrapper3, v0.2.11); PyRCN v0.3.0 was used for ELMs under Python 3.13.0. All experiments were run on a workstation with Intel Xeon E5-2680 v4 CPU, 64 GB RAM, and Ubuntu 22.04 LTS—no GPU acceleration was used, as none of the models required it.

### 2.5. Reproducibility Statement

All data sources, software versions, preprocessing scripts, model configurations, and evaluation protocols used in this study are fully documented below. The entire computational workflow—from raw data ingestion to final model evaluation—was implemented in open-source environments and can be replicated using the specifications provided. Code and processed datasets (where permissible under data-sharing agreements) are available upon request to support independent validation.

### 2.6. Code and Platform Availability

To promote transparency, reproducibility, and community reuse, the complete source code of the prediction platform developed in this study is publicly available under the GNU General Public License v3.0 (GPL-3.0) at the following GitHub 3.18 repository: https://github.com/wellingtonpinheiro/Plataforma-Predicao-Arboviroses (accessed on 10 November 2025).

The repository includes: (i) preprocessing scripts for epidemiological, entomological, and climatic data; (ii) model training and evaluation pipelines for all tested algorithms (RF, MLP, SVM, ELM, and linear regression); (iii) configuration files detailing software dependencies (see ‘requirements.txt’); and (iv) instructions for local deployment using a Python virtual environment. All code is documented with inline comments and follows modular design principles to facilitate adaptation to other geographic settings. The platform is built using open-source libraries (Python 3.13.0, scikit-learn, PyRCN, Weka, R/terra) and does not rely on proprietary tools.

## 3. Results

### Arboviruses Cases Prediction

The performance results of the models generated for predicting arbovirus cases are detailed in Table 1.

The model created with linear regression performed poorly in all metrics, with an average MAE of 0.646, average RMSE of 1.12, average RRSE of 36.857% and average correlation coefficient of 0.928, with an average training time of 0.044 s.

The models created with RF showed good results overall. With MAE ranging from 0.191 to 0.227, RMSE from 0.454 to 0.530, RRSE from 15.226% to 18.005% and correlation coefficient from 0.984 to 0.990, with training time from 0.223 to 1.761 s. The configuration with 40 trees had the best performance, with MAE of 0.118, RMSE of 0.454, RRSE% of 15.492% and correlation of 0.990, with a training time of 0.852 s, all averages of the metrics (Table 2).

Table 3 summarizes the performance of SVM models using different kernel configurations to predict *Aedes aegypti* breeding sites. The results demonstrate significant variation in predictive performance based on the kernel type and its complexity. Notably, polynomial kernels of degree 2 and 3 achieved near-perfect results, with the degree-3 configuration reaching a mean correlation coefficient of 1.000 and an RRSE of only 1.834%, indicating extremely accurate predictions. However, this performance came with a high computational cost, as training times increased sharply with model complexity—up to 1423.4 s for the cubic polynomial kernel. In contrast, the linear kernel (polynomial degree 1) and RBF kernel exhibited weaker performance, with correlation coefficients of 0.907 and 0.882, respectively, and substantially higher errors. These results underscore a trade-off between predictive accuracy and computational efficiency, suggesting that while complex kernels can capture intricate nonlinear relationships, their practical application may be limited in real-time public health settings.

The models created with MLP showed very good results in all configurations, and the average performance ranged from 0.097 to 0.114 for MAE, 0.156 to 0.181 for RMSE, 5.220% to 6.084% for RRSE% and 0.998 to 0.999 for correlation coefficient, with performance improving as the number of neurons increased. However, the training time varies from 69.651 to 124.706 s as the number of neurons increases (Table 4).

The Reservoir Computation Models and ELM produced highly variable outcomes, achieving optimal results with a single layer and the poorest performance when ten layers were used. The average performance variation ranged from 0.002 to 0.119 MAE, RMSE from 0.004 to 0.165, RRSE% from 2.386% to 101.475% and correlation coefficient from 0.298 to 0.999, with training time ranging from 0.268 to 4.599 s. The worst result was for a configuration with 10 layers and 1000 neurons in each layer. The best performance was achieved with a single layer with 1000 neurons, with the same correlation coefficient as the single layer with 900 neurons, but being slightly better for the other metrics and the training time (Table 5).

These results show the superiority of the models created with the single-layer reservoir computing paradigm. They obtained high correlation, low values for the location metrics and much shorter training times. This demonstrates the good computational cost ratio of these models for predicting *Aedes aegypti* breeding sites in the city of Recife.

## 4. Discussion

### 4.1. Implications of Model Performance for Public Health Forecasting

The present study included a large number of *Aedes aegypti* breeding sites with the aim of developing Intelligent Systems for the Prediction of arboviruses. The models created with RF presented good results, consistent with a recent study [17] which showed that the set of selected variables resulted in the best performances of the RF models to predict the number of all potential *Aedes aegypti* breeding sites. However, it highlights that since the presence of *larvae* depends on many factors, predicting only positive breeding sites may result in underestimating the real carrying capacity of the environment, highlighting the difficulty of identifying relevant predictive variables for all types of breeding sites.

This study has certain limitations. Arbovirus case data from official notifications may be underreported, misclassified, or delayed. While LIRAa is a standardized method across Brazil, its spatial and operational limits may reduce neighborhood-level detail. Despite these constraints, our decision to model both entomological indices (breeding sites) and arbovirus cases was deliberate. Entomological data provide early environmental signals of vector proliferation, while case data reflect confirmed human transmission. Combining both enhances the robustness of our predictive framework and supports more effective public health decision-making, particularly in identifying high-risk areas before outbreaks occur. The strong performance of our models, especially those based on Extreme Learning Machines and Multilayer Perceptrons, reinforces the feasibility of this approach.

Linear regression was included as a baseline model to benchmark the performance of more advanced machine learning methods. Its relatively poor predictive power, as evidenced by higher error metrics and lower correlation values, reflects its inability to capture the nonlinear and complex interactions among climate variables, entomological indicators, and disease incidence. This result is consistent with previous findings in environmental epidemiology, where linear models often fall short when applied to high-dimensional, spatiotemporal datasets. These findings reinforce the need for more sophisticated modeling approaches, such as RF, SVMs, and neural networks. These models are better suited to capturing the inherent complexity of arbovirus transmission dynamics.

The efficiency of RF for prediction applied to the spatiotemporal distribution or abundance of *Aedes aegypti* has also been demonstrated [17,18]. Randon Forest can create a highly accurate classifier using a large number of features [18,19]. In a study to identify dengue, malaria, and leishmaniasis breeding sites using spatiotemporal data in a real-time platform, Javaid [19] demonstrated that RF achieved the highest accuracies on all remaining data samples and was the best model for large datasets. Rahman [18] conducted in Northeastern Thailand on the spatial distribution and prediction of dengue vector abundance using machine learning models on different data factors such as logistic regression, SVM, k-nearest neighbor, artificial neural network, and RF, they demonstrated that RF achieves the best model performance with all data variables used in this study.

The results in Table 3 reveal a stark contrast in predictive performance across different SVM kernel types and polynomial degrees. Models employing polynomial kernels of degree 2 and 3 achieved near-perfect accuracy, with the degree-3 configuration yielding a mean correlation coefficient of 1.000, an MAE of 0.033, and an RRSE of just 1.834%. These metrics indicate an exceptionally close alignment between predicted and observed breeding site densities. However, this high predictive fidelity came at a substantial computational cost: training times escalated dramatically with kernel complexity—reaching an average of 1423.4 s for the cubic polynomial kernel, over 37 times longer than the linear variant. In contrast, the RBF kernel performed poorly (correlation = 0.882, RRSE = 48.95%), suggesting that the underlying data structure may not conform well to the isotropic similarity assumptions of RBF in this spatial context.

These findings underscore a critical trade-off between model accuracy and computational efficiency in public health applications. While high-degree polynomial SVMs can capture complex nonlinear relationships between climate variables and entomological indicators, their prolonged training times may hinder operational deployment in resource-constrained settings like municipal vector control programs. Moreover, the risk of overfitting with high-degree polynomials—especially given the limited temporal span of the breeding site data (2009–2017) warrants caution. The near-perfect performance metrics, though impressive, should be interpreted alongside robust validation on out-of-sample periods and spatial cross-validation to ensure generalizability. Thus, despite their theoretical appeal, high-degree SVMs may be less practical than lighter, more scalable alternatives for real-time arbovirus risk forecasting in dynamic urban environments such as Recife.

The results in Table 4 demonstrate that MLP models achieved consistently high predictive accuracy across all tested configurations. As the number of neurons in the hidden layer increased from 10 to 30, performance metrics improved incrementally: the MAE decreased from 0.114 to 0.097, the RRSE dropped from 6.084% to 5.220%, and the correlation coefficient remained near-perfect, ranging from 0.998 to 0.999. These findings indicate that MLPs effectively captured the nonlinear relationships between climatic variables, historical entomological data, and breeding site densities across Recife’s neighborhoods. The high correlation values, in particular, suggest that the spatial patterns predicted by the MLP closely mirrored the observed distribution of breeding sites, reinforcing the model’s suitability for fine-scale risk mapping.

However, this performance came at a notable computational cost. Training times increased substantially with model complexity—rising from approximately 69.7 s (10 neurons) to 124.7 s (30 neurons)—reflecting the iterative nature of backpropagation and the sensitivity of MLPs to hyperparameter tuning. While these durations are manageable in offline forecasting scenarios, they may pose challenges for real-time or near-real-time public health decision-making, especially in resource-limited municipal settings. Moreover, given the relatively modest gains in accuracy beyond 20 neurons, the marginal benefit of increasing model complexity must be weighed against practical deployment constraints. Nonetheless, the MLP’s strong performance underscores the value of deep learning approaches in entomological forecasting, particularly when paired with high-resolution spatiotemporal data and robust validation protocols such as the 30-fold cross-validation employed in this study.

Table 5 shows a striking sensitivity of ELM performance to architectural depth. Single-layer ELMs achieved exceptional predictive accuracy, with configurations using 900–1000 neurons yielding near-perfect correlation coefficients (0.999), extremely low MAE (≈0.001), and RRSE values below 2.4%. Training times remained remarkably short, under 0.5 s, highlighting the computational efficiency of shallow ELMs. In stark contrast, increasing the number of layers led to a rapid and consistent degradation in performance: models with 5 or 10 layers exhibited sharply reduced correlations (as low as 0.298), RRSE values exceeding 97%, and substantially higher prediction errors. This pattern indicates that, in this particular spatiotemporal forecasting setting, deeper ELM architectures experience instability, limited generalization, and inefficient random weight initialization. These problems are not solved by simply increasing their representational capacity.

These findings strongly support the use of shallow, single-layer ELMs as a high-performance, low-cost solution for operational entomological forecasting in urban public health settings. The combination of near-perfect predictive fidelity, minimal computational overhead, and robustness across repetitions positions single-layer ELMs as particularly well-suited for integration into real-time surveillance dashboards, such as those used by Recife’s vector control program. Moreover, the sharp performance drop with added layers underscores a critical insight for reservoir computing applications in epidemiology: architectural simplicity often outperforms complexity when data are spatially aggregated, temporally limited, or subject to reporting biases. Single-layer ELMs offer an optimal balance among models tested and provide a scalable, maintainable solution for municipal health systems with limited resources facing increasing climate-related arbovirus risks.

### 4.2. Comparative Analysis of Top-Performing Model Configurations

To clarify why the single-layer ELM with 1000 neurons emerged as the optimal model for predicting *Aedes aegypti* breeding sites in Recife, we directly compare its performance against the best configurations of the other four model families tested in this study.

The top RF model (40 trees) achieved strong predictive accuracy (MAE = 0.118, correlation = 0.990) and required less than 1 s of training time. However, its error metrics were over 100 times higher than those of the best ELM (MAE ≈ 0.001), indicating significantly coarser spatial resolution in risk prediction.

The Multilayer Perceptron with 30 neurons delivered near-perfect correlation (0.999) and low error (MAE = 0.097), but at the cost of prolonged training (≈125 s)—over 350 times slower than the ELM (0.34 s). This computational burden limits its suitability for real-time or frequent retraining scenarios in resource-constrained municipal health departments.

The degree-3 polynomial SVM achieved a correlation of 1.000 and an MAE of 0.033, but its average training time was over 1400 s—almost 10,000 times slower than the ELM. Additionally, it showed possible overfitting due to the short time frame of the breeding site data (2009–2017) and the lack of rigorous spatial cross-validation when working with complex kernels.

Linear regression, as expected, served only as a baseline, with substantially higher errors and lower correlation (MAE = 0.646, *r* = 0.928).

In contrast, the single-layer ELM combined near-perfect statistical performance (correlation = 0.999, RRSE = 2.39%) with exceptional computational efficiency (<0.5 s training time) and architectural simplicity. This balance is critical in public health settings, where both predictive fidelity at fine spatial scales and the ability to rapidly retrain models with incoming surveillance data are essential for actionable decision-making. The failure of deeper ELM architectures (5–10 layers) further underscores that, in this spatiotemporal context—characterized by moderate data dimensionality and aggregated neighborhood-level targets—increased representational capacity does not translate into better generalization. Instead, it introduces instability, likely due to ineffective random weight initialization and limited data for robust reservoir tuning.

Therefore, the single-layer ELM is not merely “one of the good models,” but the only architecture that simultaneously satisfies the triad of accuracy, speed, and operational robustness required for deployment in Recife’s vector control program.

The drop in predictive performance as ELM depth increases from one to ten layers is consistent with reservoir computing theory and findings in spatiotemporal forecasting with limited data. In standard ELMs, only the output layer is trained; all hidden-layer weights are assigned randomly and remain fixed. While shallow ELMs (single hidden layer) benefit from universal approximation guarantees and stable pseudo-inverse solutions, deeper ELMs lack mechanisms for error backpropagation or internal weight adaptation. In tasks like breeding site prediction with moderate features and few samples, deep random projections increase noise instead of capturing useful hierarchical patterns. This behavior is consistent with recent studies showing that deep ELMs often underperform shallow counterparts in small- to medium-scale regression tasks unless paired with sophisticated initialization schemes, layer-wise pretraining, or regularization strategies that were not employed here [21,30].

Moreover, our data exhibit strong temporal smoothness and spatial autocorrelation, properties well captured by shallow, wide networks that act as nonlinear interpolators. Deeper architectures can cause unwanted nonlinear effects that harm generalization, especially when training data is limited to short timeframes (2009–2017) or coarse entomological reports like stratum-aggregated LIRAa indices. Thus, the observed pattern is likely not unique to our scenario, but reflects a broader principle in reservoir computing: architectural depth only improves performance when the data complexity, volume, and signal-to-noise ratio justify it. In public health contexts characterized by data sparsity, reporting delays, and operational constraints, simplicity often yields greater robustness. This supports our finding that single-layer ELMs outperform alternatives and better suit municipal arbovirus surveillance needs.

We strongly encourage public health authorities and researchers to use this open infrastructure for operational validation as new epidemiological cycles unfold.

## 5. Conclusions

This study demonstrates that machine learning-based intelligent systems can effectively forecast the spatiotemporal distribution of *Aedes aegypti* breeding sites in Recife using publicly available climate and entomological data. Among the models evaluated, single-layer ELM offered the best balance of predictive accuracy—achieving correlation coefficients near 1.0 and RRSE below 2.4%—and computational efficiency, with training times under half a second. These characteristics make ELMs particularly suitable for integration into municipal surveillance platforms requiring rapid, scalable risk assessments.

While case prediction models (e.g., RF) also performed well, the strongest results were obtained for breeding site forecasting, highlighting the value of entomological indicators as early proxies for arbovirus transmission risk. The resulting fine-scale risk maps can support proactive vector control by identifying priority neighborhoods weeks in advance of peak transmission periods. Future work should incorporate socioeconomic, urban infrastructure, and real-time mobility data to enhance generalizability and extend predictions to clinical case incidence. Nonetheless, this study provides a reproducible, open-data framework that can be adapted to other tropical cities facing similar climate-driven arboviral threats.

## Figures and Tables

**Figure 1 ijerph-23-00012-f001:**
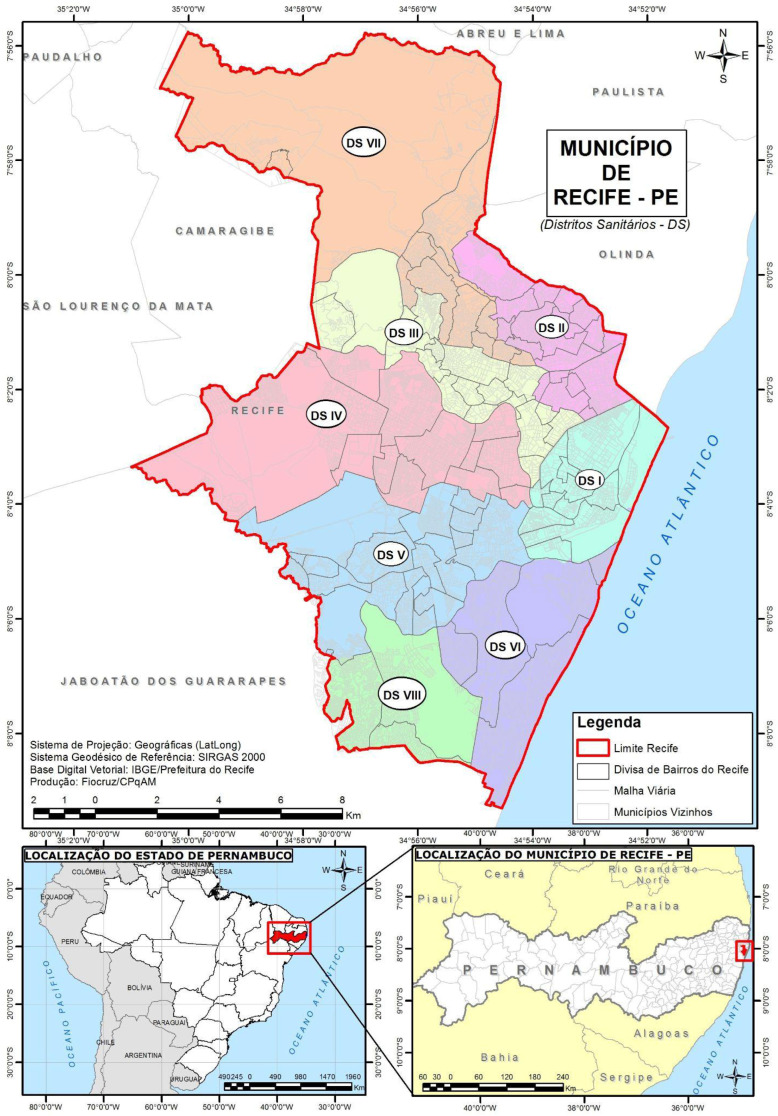
Map of the municipality of Recife (in Portuguese, *Município de Recife*), Pernambuco, identifying the eight health districts (In Portuguese, *Distritos Sanitários*: DS I, DS II, DS III, DS IV, DS V, DS VI, DS VII, and DS VIII). The boundaries of Recife are shown in red. Recife, Pernambuco, Brazil, 2020.

**Figure 2 ijerph-23-00012-f002:**
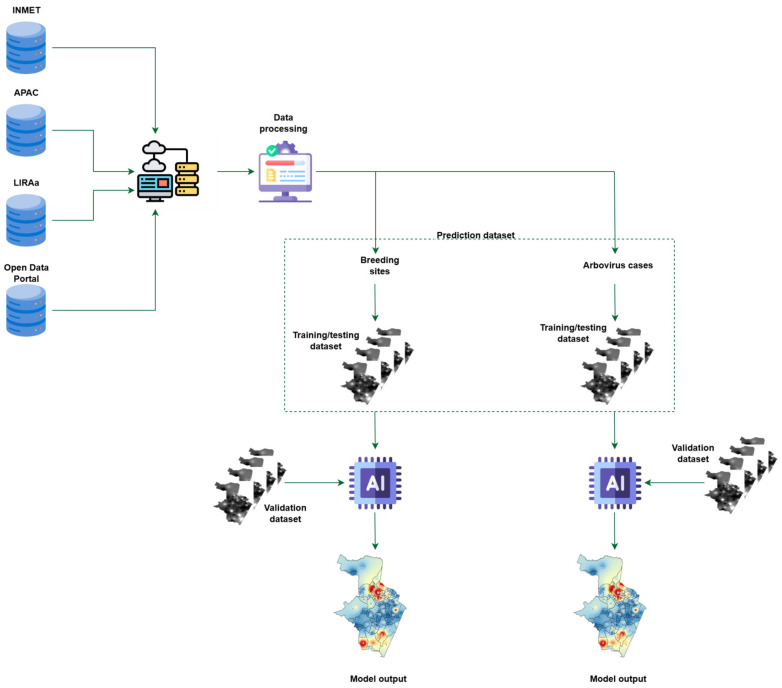
Proposed method to predict the spatial distribution of arbovirus cases in Recife, and *Aedes aegypti* breeding sites.

**Table 1 ijerph-23-00012-t001:** Prediction results for the number of breeding sites using the Linear Regression model.

	MAE		RMSE		RRSE (%)		Correlation Coefficient		Training Time (s)	
Regression Method	Mean	Std	Mean	Std	Mean	Std	Mean	Std	Mean	Std
**Linear** **Regression**	0.646	0.24	1.12	0.416	36.857	6.518	0.928	0.026	0.044	0.022

**Table 2 ijerph-23-00012-t002:** Prediction results for the number of breeding sites using RF models.

		MAE		RMSE		RRSE (%)		Correlation Coefficient		Training Time (s)	
Regression Method	Setup	Mean	Std	Mean	Std	Mean	Std	Mean	Std	Mean	Std
**Random Forest**	**10 trees**	0.227	0.078	0.530	0.215	18.005	4.109	0.984	0.008	0.223	0.085
	**20 trees**	0.203	0.070	0.480	0.199	16.313	3.944	0.988	0.007	0.382	0.122
	**30 trees**	0.191	0.064	0.459	0.192	15.619	3.937	0.989	0.007	0.867	0.699
	**40 trees**	0.118	0.065	0.454	0.192	15.421	3.925	0.989	0.007	0.852	0.170
	**50 trees**	0.216	0.067	0.521	0.205	15.627	3.703	0.989	0.006	1.007	0.140
	**60 trees**	0.213	0.066	0.517	0.204	15.490	3.703	0.990	0.006	0.956	0.146
	**70 trees**	0.211	0.066	0.514	0.204	15.405	3.711	0.990	0.006	1.250	0.311
	**80 trees**	0.209	0.065	0.511	0.203	15.333	3.713	0.990	0.006	1.621	0.238
	**100 trees**	0.207	0.065	0.508	0.203	15.226	3.704	0.990	0.006	1.761	0.257

**Table 3 ijerph-23-00012-t003:** Prediction results for the number of breeding sites using Support Vector Machine models.

		MAE		RMSE		RRSE (%)		Correlation Coefficient		Training Time (s)	
Regression Method	Setup	Mean	Std	Mean	Std	Mean	Std	Mean	Std	Mean	Std
**SVM**	**polynomial kernel, *p* = 1**	0.514	0.211	1.231	0.560	41.243	10.357	0.907	0.044	38.135	6.673
	**polynomial kernel, *p* = 2**	0.057	0.023	0.123	0.068	4.171	2.052	0.999	0.002	392.845	197.922
	**polynomial kernel, *p* = 3**	0.033	0.013	0.055	0.046	1.834	1.377	1.000	0.001	1423.400	838.156
	**polynomial kernel, RBF**	0.715	0.271	1.454	0.625	48.951	8.715	0.882	0.045	50.354	16.332

**Table 4 ijerph-23-00012-t004:** Prediction results for the number of breeding sites using Multilayer Perceptron models.

		MAE		RMSE		RRSE (%)		Correlation Coefficient		Training Time (s)	
Regression Method	Setup	Mean	Std	Mean	Std	Mean	Std	Mean	Std	Mean	Std
**MLP**	**10 neurons**	0.114	0.065	0.181	0.104	6.084	2.648	0.998	0.002	69.651	31.176
	**20 neurons**	0.106	0.062	0.168	0.101	5.630	2.561	0.999	0.002	111.492	36.160
	**30 neurons**	0.097	0.056	0.156	0.097	5.220	2.533	0.999	0.002	124.706	34.655

**Table 5 ijerph-23-00012-t005:** Prediction results for the number of breeding sites using Extreme Learning Machine models.

			MAE		RMSE		RRSE (%)		Correlation Coefficient		Training Time (s)	
Regression Method	Setup	Neurons	Mean	Std	Mean	Std	Mean	Std	Mean	Std	Mean	Std
**ELM**	**1 layers**	500	0.002	0.001	0.007	0.006	4.375	3.463	0.998	0.004	0.268	0.168
		700	0.001	0.001	0.006	0.007	3.366	3.789	0.999	0.005	0.338	0.200
		900	0.001	0.000	0.004	0.005	2.575	2.846	0.999	0.002	0.405	0.215
		1000	0.001	0.000	0.004	0.005	2.386	2.717	0.999	0.002	0.339	0.135
	**2 layers**	500	0.021	0.004	0.037	0.015	22.570	9.005	0.973	0.024	0.401	0.162
		700	0.017	0.005	0.036	0.071	22.510	46.623	0.975	0.059	0.557	0.305
		900	0.015	0.003	0.030	0.030	18.237	18.225	0.982	0.044	0.700	0.259
		1000	0.014	0.003	0.032	0.037	19.810	22.472	0.977	0.055	0.903	1.867
	**5 layers**	500	0.073	0.012	0.104	0.017	63.867	5.068	0.773	0.039	0.887	0.254
		700	0.073	0.011	0.101	0.016	62.498	5.181	0.785	0.037	2.180	64.080
		900	0.071	0.011	0.098	0.015	60.249	5.031	0.804	0.034	2.760	77.216
		1000	0.070	0.011	0.096	0.016	59.154	5.060	0.812	0.033	2.183	11.042
	**10 layers**	500	0.111	0.020	0.160	0.027	97.911	3.102	0.299	0.061	1.711	0.293
		700	0.115	0.020	0.162	0.027	99.447	3.684	0.298	0.066	3.855	2.446
		900	0.117	0.020	0.162	0.027	99.753	4.122	0.328	0.062	4.097	0.574
		1000	0.119	0.021	0.165	0.027	101.475	4.287	0.312	0.062	4.599	0.712

## Data Availability

The original data presented in the study are openly available in APAC Geographic Information System (SIGH-PE) https://www.apac.pe.gov.br (accessed on 10 November 2025); INMET Meteorological Database for Teaching and Research (BDMEP) https://bdmep.inmet.gov.br (accessed on 10 November 2025); and The Open Data Portal of Recife https://dados.recife.pe.gov.br (accessed on 10 November 2025).

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
