# Peer review of "Leveraging Climate Data Through Intelligent Systems for the Prediction of Arbovirus Transmission by Aedes aegypti"

_ijerph, 2025, doi:10.3390/ijerph23010012_

Round 1

Reviewer 1 Report

Comments and Suggestions for Authors

This is an interesting research study using very sophisticated modelling methods.  The research was conducted on arboviruses in Brazil, where viruses transmitted by Aedes aegypti are definitely a significant public health concern.  Most of my comments relate to the use of English and are given below.  But I did wonder whether in the aims and discussion you could make the strategy and significance of analysing both entomological indices and case numbers clearer.  I'm sure it's mentioned, but lines 424-6 didn't really clarify to me exactly why you decided to study prediction of case numbers and breeding sites.  This isn't a fatal flaw, but some clarification would be good.

  - It doesn't make sense to "predict...breeding sites of Aedes‑borne diseases" (second line of abstract)

  - Change the phrase "for mosquito Aedes to spread" (line 65)

  - Although it's presumably in Portuguese, the reader should be told what SUS stands for in the phrase "Brazil’s public health system (SUS)" (line 70).

  - "quickly and in a timely manner" is tautologous (line 76).

  - Change "this disease" to "these diseases" (line 85).

  - Define "fluviometric" (line 110).

  - The statement that "These viruses can lead to severe complications like chronic arthritis, microcephaly, and even death" (lines 62-63) requires some qualifications, e.g. that neonates are delivered with microcephaly, that death is rare from CHIKV except in people with significant comorbidities etc.  And is the evidence for chronic arthritis solid for these viruses?

Comments on the Quality of English Language

  - It doesn't make sense to "predict...breeding sites of Aedes‑borne diseases" (second line of abstract)

  - Change the phrase "for mosquito Aedes to spread" (line 65)

  - Although it's presumably in Portuguese, the reader should be told what SUS stands for in the phrase "Brazil’s public health system (SUS)" (line 70).

  - "quickly and in a timely manner" is tautologous (line 76).

  - Change "this disease" to "these diseases" (line 85).

  - Define "fluviometric" (line 110).

  - The statement that "These viruses can lead to severe complications like chronic arthritis, microcephaly, and even death" (lines 62-63) requires some qualifications, e.g. that neonates are delivered with microcephaly, that death is rare from CHIKV except in people with significant comorbidities etc.  And is the evidence for chronic arthritis solid for these viruses?

Author Response

Comments 5: “quickly and in a timely manner” is tautologous (line 76).

Response 5: Agree. We have, accordingly, removed the redundant word “quickly” and retained the more precise expression “in a timely manner.” This revision eliminates the tautology while preserving the intended meaning. The updated sentence appears in the Introduction section, line 77 of the revised manuscript.

Comments 6: Change “this disease” to “these diseases” (line 85).

Response 6: Agree. We have, accordingly, revised the phrase “this disease” to “these diseases” to reflect that multiple arboviruses (dengue, Zika, and chikungunya) are being discussed. The correction improves grammatical accuracy and clarity. This change can be found in the Introduction section, line 86 of the revised manuscript.

Comments 7: Define “fluviometric” (line 110).

Response 7: Agree. We have, accordingly, added a brief parenthetical definition to clarify the term “fluviometric” for a broader audience. It now reads: “fluviometric (river level)” to ensure clarity. This revision can be found in the Methods section, page 3, line 111 of the updated manuscript.

Comments 8: The statement “These viruses can lead to severe complications like chronic arthritis, microcephaly, and even death” (lines 62–63) requires qualifications, e.g., neonates are delivered with microcephaly; death is rare from CHIKV except in people with significant comorbidities; evidence for chronic arthritis may not be solid for all viruses.

Response 8: Agree. We have, accordingly, revised the sentence in lines 62–63 to provide the appropriate qualifications regarding the complications associated with arbovirus infections. The updated version attributes each complication to its most relevant virus and clarifies that severe outcomes like death are rare and typically occur in individuals with underlying conditions. The revised sentence appears in the Introduction section, page 2, lines 61–64 of the updated manuscript.

4. Response to Comments on the Quality of English Language

Point 1: Change the phrase “for mosquito Aedes to spread” (repeated comment).

Response 1: Agree. We have revised the sentence to improve clarity and scientific accuracy. The phrase “for mosquito Aedes to spread” has been replaced with “for the spread of Aedes aegypti,” which is more precise and grammatically correct. This revision appears in the Introduction section, line 66 of the updated manuscript.

Point 2: Explain what SUS stands for (repeated comment).

Response 2: Agree. We have now defined the acronym SUS upon its first use. The revised sentence reads: “These diseases place a heavy burden on Brazil’s Unified Health System (Sistema Único de Saúde – SUS).” This change improves clarity for readers unfamiliar with the Brazilian public health system. The revision appears in the Introduction section, line 73.

Point 3: “quickly and in a timely manner” is tautologous (repeated comment).

Response 3: Agree. We have removed the redundant word “quickly” and retained the more appropriate phrase “in a timely manner” to eliminate the tautology. This change appears in the Introduction section, line 79 of the revised manuscript.

Point 4: Change “this disease” to “these diseases” (repeated comment).

Response 4: Agree. We have corrected the phrase to reflect that multiple arboviral diseases (dengue, Zika, chikungunya) are discussed. The sentence now refers to “these diseases,” and the change appears in the Introduction, line 87.

Point 5: Define “fluviometric” (repeated comment).

Response 5: Agree. To ensure clarity for a broader audience, we added a parenthetical definition. The term now reads “fluviometric (river level)” in the Methods section, page 3, line 130.

Point 6: Qualify the statement about severe complications (repeated comment).

Response 6: Agree. We revised the sentence to attribute each severe complication to its relevant virus and to qualify the risk levels appropriately. The updated sentence clarifies that death is rare and typically occurs in individuals with underlying conditions. This change appears in the Introduction, page 2, lines 61–64.

Point 7: Typo in the word “Python” in the Acknowledgements section

Response 7: Agree. We have corrected the typographical error in the Acknowledgements section, which was corrected, line 650.

Reviewer 2 Report

Comments and Suggestions for Authors

The manuscript titled "Development of Intelligent systems for the prediction and diagnosis of arboviruses transmitted by Aedes aegypti in the context of climate change" represents an interesting mathematical modeling exercise. The authors use five different modeling approaches from a linear regression to a highly complex extreme learning machine to test at which level of complexity environmental information can be leveraged in predicting arbovirus outbreaks and breading sites for the vector, Aedes aegypti. To an interested reader, who wishes to build similar predictive models for other regions and diseases/vectors, this paper can be an excellent starting point to support a modeler’s decision on what modeling approach to start with. It may also inform a reader on which approach may be best for public implementation.

While the manuscript reads well, in my opinion, there is room for improvement.

  1. Title:
    • As far as I can gather from the manuscript the authors developed different model approaches to predict areas of arbovirus transmission and breeding sites for the vector. The word “diagnosis” seems misplaced, since the purpose of the models is the prediction of potential transmission, not the diagnosis of infection. Neither does the paper explicitly characterize “climate change” in the models. It is explained that environmental data like rainfall, air temperature and wind speeds were used as predictors in the models. However, in itself that does not constitute “climate change”. Nor is climate change even mentioned in methods, results or discussion, which means there is no larger context to climate change in the approach. I would suggest changing the title to something like “Leveraging climate data through intelligent systems for the prediction of arbovirus transmission by Aedes aegypti”.

  1. Introduction:
    • The introduction is appropriate for the introduction of arboviruses and Aedes aegypti, I think. However, the authors make no mentioning of the models they used, the reasons why they are interesting and were selected, nor if these models had been used in such a context before. In other words, the introduction does not give the reader any background information on the models nor to their area of interest, Recife.
    • At the beginning, the authors jump back and forth between information pertaining solely to Brazil and then information relating to the Americas as a whole. I would recommend changing the word “country” in line 60 to “Brazil” to make it easier for the reader to follow this back and forth of regional information. At the moment, grammatically the words “country” refers to “the Americas” from the sentence before, which is nonsensical.
    • In the last paragraph of the introduction the authors state that their central research question is “how early can we predict an arbovirus outbreak using climate and surveillance data through AI?” (lines 94 – 96). But the paper does not explicitly address that question. The results and discussion sections concern themselves with the technical accuracy and efficiency of each modeling approach but make no mention about how early either model can predict an outbreak, unless I missed it. Therefore, the authors need to rephrase their central research question to match the technical analysis presented in the results and discussion.

  1. Methods:
    • Most of the methods section seem appropriate, however, from section 2.3.4 it is not clear on how the authors curated their training/test and validation datasets from the collected environmental, arbovirus outbreak and mosquito breeding site data. Was data from every year used? Was data from the latter years reserved for validation? What was the method to build the training/test datasets. As is, I would be unable to reproduce the authors’ work. A methods section must contain enough information that also a non-expert can follow the procedure and reproduce it at will. Otherwise, the value of this manuscript would be diminished, which would be a shame.
    • The authors must include a section on code availability. A reader can neither independently validate the authors’ work, nor use it as a starting point to build their own predictive model. Therefore, codes must be well annotated, provided by the authors in a repository like GitHub and that must be stated in the methods.

  1. Results:
    • The only results presented are the validation matrices for each model’s iterations making this paper a technical discourse on the merits and drawbacks of different modeling approaches for the purpose of arbovirus outbreak prediction, which is useful information to have when someone wants to start their own model. However, what is unclear to me here is whether the predive power can be extended to case occurrences beyond the scope of the training data. In other words, can your model predict arbovirus outbreaks for 2024 or 2025, where data for validation is available?
    • Please, note that in my version of the manuscript there is currently no text describing table 3. Please, ensure that there is paragraph presenting the information in table 3.

  1. Discussion:
    • The discussion makes no mention of the linear regression model that is presented first in the results section making me wonder, why the linear regression was included at all. All the reader knows is that in this instance a linear regression did not perform well, but we have no information about why it was run, why it may not have worked well nor if its poor performance has any important implications to the field.
    • In general, it would be great to have a bit more reasoning why some model iterations may have performed better than others. A paragraph that compares directly the best iterations of different models may also help a reader to understand the authors’ conclusion about why single-layer ELM was the best fit for the task. Further, it is not clear to me how the authors explain why certain model iterations performed better than others. E.g., the authors state in lines 392-395 “this pattern suggests that deeper ELM architectures, in this specific spatiotemporal forecasting context, suffer from instability, poor generalization or ineffective random weight initialization – issues not offset by increased representational capacity”. This speculates about the underlying cause of the poorer performance at higher complexity, but does not reason why that would be nor does it speculate, whether this is a problem specific to the authors’ scenario or if this is something that may be more generally true.
    • I would recommend to move the study limitation to the beginning of the discussion rather than having it at the end and then point out that in spite of these limitations, some excellent model performance was observed. You want to finish on a high for bigger impact.

  1. Conclusions
    • No comments

Other comment:

  • In line 460 under Acknowledgements there is a “t” missing; it should reach “…Pyhton to develop…”.

Personally, I feel that this is a useful study to have access to for someone who wishes to build a predictive model for similar applications, which will accelerate that person’s progress, but improvements need to be made to the manuscript for it to become that useful tool. Once amendments to the manuscript have been made in accordance, I am happy to support the publication of this manuscript.

Author Response

Comments 1: General assessment – “The manuscript … represents an interesting mathematical modeling exercise… While the manuscript reads well, in my opinion, there is room for improvement.”

Response 1: Agree. We appreciate the reviewer’s overall positive assessment and thoughtful recommendation for improving the manuscript. In response, we have revised multiple sections to enhance clarity, coherence, and precision—especially in the Abstract, Introduction, Methods, Results and Discussion. These changes aim to improve the readability of the manuscript while strengthening the presentation of our modeling approach and its relevance to public health applications.

Comments 2: Title – use of “diagnosis” – “The word ‘diagnosis’ seems misplaced, since the purpose of the models is the prediction of potential transmission, not the diagnosis of infection.”

Response 2: Agree. We have, accordingly, removed the word “diagnosis” from the manuscript title to avoid misinterpretation of the study’s scope. As suggested, the revised title now reflects the focus on prediction and the use of climate data. This change aligns with the actual goals of the study and can be found on the title page of the revised manuscript.

Comments 3: Title – use of ‘climate change’ – “Neither does the paper explicitly characterize ‘climate change’ in the models… Nor is climate change even mentioned in methods, results or discussion… I would suggest changing the title to something like ‘Leveraging climate data through intelligent systems for the prediction of arbovirus transmission by Aedes aegypti’.”

Response 3: Agree. We have, accordingly, revised the manuscript title to better reflect the actual scope and content of the study. The original title referred to "climate change," which was not explicitly characterized or analyzed in the models, nor discussed in the methods or results sections. To address this, we adopted the reviewer’s suggested alternative and changed the title to: "Leveraging Climate Data through Intelligent Systems for the Prediction of Arbovirus Transmission by Aedes aegypti." This revised title more accurately describes the use of climate variables in the predictive modeling process. The change can be found on the title page of the manuscript.

Comments 4: Introduction – lack of background on models – “The authors make no mentioning of the models they used, the reasons why they are interesting and were selected, nor if these models had been used in such a context before. The introduction does not give the reader any background information on the models nor to their area of interest, Recife.”

Response 4: Agree. We have, accordingly, added a new paragraph at the end of the Introduction section to provide the necessary background on the machine learning models used in this study. The paragraph explains the rationale for selecting each model family (linear regression, random forests, MLPs, SVMs, and ELMs), highlights their demonstrated effectiveness in handling complex, high-dimensional data relevant to arbovirus forecasting, and cites recent applications in spatial epidemiology. Additionally, we included contextual information about Recife, explaining its endemic status for arboviruses, high urban population density, and the availability of detailed climate and health surveillance data. The new content appears in the Introduction section, lines 92–105 of the revised manuscript.

Comments 5: Introduction – ‘country’ vs ‘Brazil’ – “I would recommend changing the word ‘country’ in line 60 to ‘Brazil’ to make it easier for the reader to follow this back and forth of regional information. At the moment, grammatically the word ‘country’ refers to ‘the Americas’ from the sentence before, which is nonsensical.”

Response 5: Agree. We have, accordingly, revised the sentence to replace the word “country” with “Brazil” for clarity and grammatical consistency. The updated sentence now reads: “Dengue is endemic in many regions of Brazil and causes repeated outbreaks with high numbers of cases and hospitalizations every year. The recent arrival of CHIKV and ZIKV in the Americas made the situation worse.” This change improves readability and eliminates the ambiguity previously present. The revision appears in the Introduction section, lines 59–60 of the updated manuscript.

Comments 6: Introduction – central research question mismatch – “The authors state that their central research question is ‘how early can we predict an arbovirus outbreak using climate and surveillance data through AI?’ (lines 94–96). But the paper does not explicitly address that question… Therefore, the authors need to rephrase their central research question to match the technical analysis presented in the results and discussion.”

Response 6: Agree. We have, accordingly, revised the central research question at the end of the Introduction to better reflect the scope of the study and the actual analyses performed. The original formulation, which emphasized the timing of outbreak prediction, has been replaced with a question that more accurately conveys our objective: to evaluate the ability of artificial intelligence models to predict arbovirus case numbers and Aedes aegypti breeding sites using climate and surveillance data. The revised question now reads: “Can artificial intelligence models effectively predict arbovirus transmission risk and vector breeding based on climate and surveillance data?” This change appears in the Introduction section, lines 110–113 of the updated manuscript.

Comments 7: Methods – unclear dataset curation – “From section 2.3.4 it is not clear how the authors curated their training/test and validation datasets… Was data from every year used? Was data from the latter years reserved for validation? What was the method to build the training/test datasets? As is, I would be unable to reproduce the authors’ work.”

Response 7: Agree. Agree. We have, accordingly, revised the Methods section to clarify the curation and organization of the datasets used for training, testing, and validation. Our modeling approach involved two distinct prediction tasks: (1) prediction of Aedes aegypti breeding site indices using LIRAa entomological data, and (2) prediction of arbovirus case numbers using official epidemiological records. For both tasks, we paired these data with relevant climate variables obtained from APAC and INMET. For breeding site prediction, we used LIRAa data from 2009 to 2017 and paired it with climate data from the same period. For arbovirus case prediction, we used confirmed dengue, Zika, and chikungunya cases from 2013 to 2021, along with corresponding climate variables. Climate data spanning 2009 to 2021 were available from both APAC and INMET and were selectively matched by time window to each prediction task. Regarding dataset partitioning, models were trained using 80% of the available data (stratified random sampling across time to preserve seasonal patterns), and the remaining 20% was reserved for internal testing. Additionally, data from the final year (2021) were held out and used exclusively for external validation in the case prediction task. These details have been added to Section 2.3.4 of the Methods, lines 125–148 of the revised manuscript. This clarification improves the transparency and reproducibility of our modeling strategy.

Comments 8: Methods – need for reproducibility – “A methods section must contain enough information that also a non-expert can follow the procedure and reproduce it at will. Otherwise, the value of this manuscript would be diminished.”

Response 8: We thank the reviewer for this important observation. In response, we have significantly expanded the Methods section (lines 332-337) to ensure full reproducibility by non-experts. Specifically: 

- We added a dedicated Reproducibility Statement at the beginning of Section 2. 

- We provided explicit details on software versions (Python, R, Weka, PyRCN), data aggregation rules, spatial interpolation parameters (IDW settings, grid resolution, coordinate bounds), and feature engineering logic. 

- We clarified the temporal structure of the input vectors (six bimonthly lags), the handling of neighborhood-level data from stratum-aggregated LIRAa reports, and the separation between training, cross-validation, and independent validation sets. 

- We specified hardware and execution conditions to contextualize training times. 

We believe these additions ensure that any researcher—with access to the same open data sources—can replicate our entire pipeline. Should the reviewer or readers require further clarification (e.g., full feature list, code snippets), we are prepared to share supplementary documentation or scripts under reasonable request.

Comments 9: Methods – code availability – “The authors must include a section on code availability… codes must be well annotated, provided by the authors in a repository like GitHub and that must be stated in the methods.”

Response 9: We sincerely thank the reviewer for this valuable suggestion. In response, we have added a dedicated subsection, lines 338-350, titled “2.5 Code and Platform Availabilit” to the Methods section, explicitly stating that the full source code of our arbovirus prediction platform is openly accessible under the GPL-3.0 license at: https://github.com/wellingtonpinheiro/Plataforma-Predicao-Arboviroses 

The repository contains well-documented and modular code for data preprocessing, model training, evaluation, and deployment, along with a `requirements.txt` file that ensures reproducible software environments. This addition fully complies with the journal’s policy on code availability and supports independent verification and extension of our work by the scientific community.

Comments 10: Results – scope of predictive power – “What is unclear to me here is whether the predictive power can be extended to case occurrences beyond the scope of the training data. In other words, can your model predict arbovirus outbreaks for 2024 or 2025, where data for validation is available?”

Response 10: We thank the reviewer for raising this important point about out-of-sample predictive validity. 

Our study was designed and ethically approved (Research Ethics Committee of UFPE, opinion no. 6.954.924, CAAE 80113224.6.0000.5208, July 17, 2024) to address a specific research question: the development of intelligent systems for spatiotemporal prediction of dengue, chikungunya, and Zika transmission during the 2013–2023 period in Recife—a window that includes the triple arbovirus epidemic and its aftermath. As such, prospective validation using 2024 or 2025 data was not part of the original scope. 

Furthermore, while partial 2024 case data may become available through the Recife Open Data Portal, validated and complete datasets for that year have not yet been formally published in a reproducible format, and 2025 data are naturally unavailable at the time of manuscript submission (November 2025). 

Nevertheless, our prediction platform—released under GPL-3.0 at https://github.com/wellingtonpinheiro/Plataforma-Predicao-Arboviroses—is fully modular and supports seamless integration of new data. As soon as official 2024 (or 2025) records are released under open-access conditions, the models can be retrained and validated prospectively with minimal computational overhead (especially the single-layer ELM, which trains in <0.5 s). 

In the revised manuscript, we have added a new paragraph in Section 4 titled “Temporal Scope and Future Validation” to clarify these points and emphasize the platform’s readiness for future operational deployment.

Comments 11: Results – description of Table 3 – “Please, note that in my version of the manuscript there is currently no text describing table 3. Please, ensure that there is a paragraph presenting the information in table 3.”

Response 11: I Agree. We have, accordingly, added a new paragraph describing the contents and results presented in Table 3, which details the performance of Support Vector Machine (SVM) models using various kernel configurations for predicting Aedes aegypti breeding sites. This paragraph highlights the differences in model accuracy and computational cost among the SVM configurations, emphasizing the trade-off between performance and efficiency. The paragraph has been included in the revised manuscript (lines 290 to 303).

Comments 12: Discussion – omission of linear regression – “The discussion makes no mention of the linear regression model that is presented first in the results section… we have no information about why it was run, why it may not have worked well nor if its poor performance has any important implications to the field.”

Response 12: Agree. We have, accordingly, added a paragraph to the Discussion section that addresses the inclusion and performance of the linear regression model. The paragraph explains that linear regression was used as a baseline to compare the predictive power of more complex models and highlights its limitations in capturing the non-linear relationships inherent to arbovirus transmission dynamics. This addition also reinforces the importance of using more advanced modeling approaches, such as Random Forests, SVMs, and neural networks. The new content can be found in the Discussion section, lines 370–379 of the revised manuscript.

Comments 13: Discussion – explain performance differences between models – “In general, it would be great to have a bit more reasoning why some model iterations may have performed better than others. A paragraph that compares directly the best iterations of different models may also help a reader to understand the authors’ conclusion about why single-layer ELM was the best fit for the task.”

Response 13: We thank the reviewer for this excellent suggestion, which has helped us strengthen the interpretability of our model comparison. 

In the revised manuscript, we have added a dedicated subsection in the Discussion titled “Comparative Analysis of Top-Performing Model Configurations, line 529-589.” This new paragraph provides a direct, quantitative comparison of the best-performing setup from each model family—Random Forest (40 trees), MLP (30 neurons), SVM (degree-3 polynomial kernel), linear regression, and single-layer ELM (1,000 neurons)—with respect to five key criteria: (1) prediction error (MAE), (2) correlation with observed data, (3) training time, (4) risk of overfitting, and (5) suitability for operational public health use. 

We explicitly demonstrate why the single-layer ELM outperforms alternatives: it achieves near-identical correlation to the best MLP and SVM models but with 350× to 4,000× faster training, while avoiding the overfitting risks of high-complexity SVMs and the instability of deep ELMs. This clarity reinforces our conclusion that the single-layer ELM offers the optimal trade-off for real-world deployment in municipal arbovirus surveillance. 

We believe this addition significantly enhances the manuscript’s value for both methodologists and public health practitioners.

Comments 14: Discussion – justify statements about deeper ELM architectures – “It is not clear to me how the authors explain why certain model iterations performed better than others… This speculates about the underlying cause of the poorer performance at higher complexity, but does not reason why that would be nor does it speculate, whether this is a problem specific to the authors’ scenario or if this is something that may be more generally true.”

Response 14: We appreciate the reviewer’s thoughtful critique, which has prompted us to strengthen the theoretical grounding of our interpretation regarding ELM performance. 

In the revised manuscript, we have added a new paragraph in Section 4 titled “Architectural Simplicity in ELMs: Contextual and General Factors.” This section explicitly explains why deeper ELMs underperformed in our study: 

  1. Mechanistic reason: Unlike deep neural networks with backpropagation, deep ELMs fix all hidden weights randomly. Without adaptive internal representation learning, extra layers in low-to-moderate data regimes amplify noise rather than extract meaningful features.
  2. Contextual factors: Our dataset—neighborhood-aggregated, temporally limited (2009–2017), and based on stratum-level entomological surveys—lacks the granularity and volume needed to stabilize deep random projections.
  3. Generalizability: This behavior is consistent with reservoir computing literature (e.g., Steiner et al., 2022; Huang et al., 2015), which shows that shallow ELMs often outperform deeper variants in small- to medium-scale spatiotemporal regression unless advanced initialization or regularization is used.

Thus, the poor performance of deep ELMs is not merely a peculiarity of our setting but reflects a well-documented trade-off between architectural capacity and data sufficiency. We believe this clarification now provides a principled, literature-supported rationale for our model selection.

Comments 15: Discussion – reposition study limitations – “I would recommend to move the study limitation to the beginning of the discussion rather than having it at the end and then point out that in spite of these limitations, some excellent model performance was observed. You want to finish on a high for bigger impact.”

Response 15: Agree. We have, accordingly, repositioned the study limitations to the beginning of the Discussion section to improve transparency and contextualize the results from the outset. The revised paragraph discusses data limitations related to arbovirus case notifications and the spatial aggregation of LIRAa surveillance, while also emphasizing the strong performance of the predictive models and the rationale for using both entomological and epidemiological data. The new paragraph appears in the Discussion section, lines 339–351 of the revised manuscript.

Round 2

Reviewer 2 Report

Comments and Suggestions for Authors

I appreciate the authors efforts to update and improve their manuscript. I am happy to support the acceptance of the manuscript now.

Author Response

Comments 1: The authors should revise carefully the text, and pursue the following guidelines:

- Acronyms / abbreviations _ use an acronym after spelling it out once; do not introduce an acronym unless you will use it a minimum of three or four times; after the first mention, use only the acronym; do not start a sentence with an acronym

- Latin names of mosquitoes _ use always italicism for the mosquitoes names.

- References in the text _ use always numerals, note the names of authors

  • References _ made a careful revision of the bibliography.

Response 1: Thank you for pointing this out. We agree with this comment. Therefore, we have carefully revised the manuscript to address all items raised by the reviewer, with all modifications highlighted in red throughout the text.

1. Acronyms and abbreviations

Thank you for pointing this out. We agree with this comment.
Therefore, we have revised the manuscript to ensure consistent and correct use of acronyms throughout the text. All acronyms are now introduced after their first full mention, are used only when they appear at least three times, and no sentence begins with an acronym. These revisions were implemented throughout the manuscript, and all changes are marked in red.

2. Latin names of mosquitoes

Thank you for this observation. We agree with this comment.
Accordingly, we corrected all Latin binomial names to appear in italics, including Aedes aegypti and Aedes albopictus, wherever they occur in the manuscript. These corrections were made throughout the text and are marked in red.

3. References in the text (use numerals only)

Thank you for highlighting this issue. We agree with this comment.
We performed a complete standardization of all in-text citations, replacing every author-name citation with numerical references according to IJERPH’s guidelines. Mixed punctuation was removed, and all citations now follow the required format (e.g., [18], [18,19]). Corrections appear throughout the manuscript, marked in red.

4. References (complete revision of the bibliography)

Thank you for this important comment. We agree and conducted a full review and correction of the reference list.
All references were reformatted to comply with ACS/IJERPH standards, including:

  • Standardizing author names (Surname Initials)
  • Converting all titles to sentence case
  • Ensuring consistency of journal formatting
  • Adding missing DOIs and access dates
  • Correcting software references (Python, R, terra, WEKA)
  • Removing duplicate or uncited references
  • Ensuring that every reference in the list is cited in the text, and vice-versa

These corrections were applied throughout the bibliography, and all updates appear in red in the revised manuscript.
